# Forestry Scenario Modelling: Qualitative Analysis of User Needs in Lithuania

**Daiva Juknelienė** *, **Michailas Palicinas, Jolanta Valčiukienė and Gintautas Mozgeris**

Agriculture Academy, Vytautas Magnus University, Studentų Str. 11, Akademija, 53361 Kaunas Distr., Lithuania; michailas.palicinas@vmu.lt (M.P.); jolanta.valciukiene@vdu.lt (J.V.); gintautas.mozgeris@vdu.lt (G.M.)
* Correspondence: daiva.jukneliene@vdu.lt

**Abstract:** This paper aims to demonstrate the use of qualitative research methods, specifically in-depth interviews, to explore the intangible and often difficult-to-quantify needs for forestry scenario modelling in Lithuania, which are frequently not adequately perceived. The study involved informants representing key actors in forest policy, forest management, research, and education. A total of 21 informants from 11 different institutions, which hold significant power and expertise in forest decision making, were interviewed. The purpose of these interviews was to gather their perspectives on the potential forest decision support system in the country, aiming to address most of their needs. The interview questions explored various aspects, including the requirements for forestry scenario modelling, the desired level of detail and information content for decision making, and both functional and nonfunctional requirements for the scenario modelling system. It is worth noting that the expected functionality of the planned forest DSSs aligns with modern international standards. Nevertheless, the diversity of perspectives, wishes, visions, and intentions of key Lithuanian forestry actors regarding the aims, objectives, and essential functionality of forestry scenario modelling tools were identified. The understanding of the requirements for modern forest DSSs was greatly influenced by the current forestry paradigms in the country and the professional experiences of individual informants. In conclusion, our findings demonstrate that the utilization of qualitative research, particularly through in-depth interviews, has proven to be a highly effective tool for accurately specifying the requirements of a modern forest DSS. It helped mitigate preconceived notions and address gaps in the envisioned product, specifically by developing a framework of core solutions for the national forestry and land-use scenario modelling system.

**Keywords:** forestry; scenario modelling; decision support; information systems; qualitative research





## 1. Introduction

Scenario modelling is usually associated with the application of information technology; however, the human desire to look forward to the future is older than any computer. With the birth of forestry in Europe, foresters also desired to learn about the consequences of their activities in order to be able to adopt optimal decisions when they need to be made. The first examples of the application of forestry scenario modelling include the classical methods of measuring the extent of forest use developed in the 18th century, where volume increment was used as a measure to determine the cutting area [1]. Later examples of forestry model include developed yield tables or their expressions in the form of a mathematical formula used for determining the current of future growing stock volume and for substantiating forestry solutions. Currently, a number of methodological solutions are used for forestry scenario modelling: yield tables, empirical stand-level models, stochastic models, individual tree models, and models of ecophysiological processes at tree and stand level [2].

Forestry scenarios are supposed to depict and describe a range of possible, preferable, and probable future developments of forests and forest management. They are implemented using diversity of computer-driven modelling tools. Forestry scenario modelling

helps to support different forestry decisions. Often it is implemented as an analysis of scenarios which anticipates the possible consequences of choosing one or another decision. In the past, in Europe, the most common aim was to assess the potential consequences of increased use of forests, with a particular focus on the sustainability of wood supply to industry (e.g., in Sweden). When the area of forests began to steadily rise in some countries, the issue of the balance between wood production, protection of biodiversity, recreation, and other ecosystem services provided by the forest using one or another forestry system became increasingly more important [2]. Currently, taking into consideration global climate change mitigation initiatives and EU commitments to reduce the greenhouse gas emissions [3,4], the importance of forestry decisions on carbon sequestration and other ecosystem services is increasing. In parallel, the impact of climate change on forest growth and state [5] and the general significance of wood in bioeconomy needs also to be assessed.

The peculiarities of forestry scenario modelling systems are related to the peculiarities of country forests, forestry, and forest inventories. There are many detailed references summarizing the methods used for forestry scenario modelling and their implementation in forestry decision support systems [6–12]. There are also guidelines for standardized evaluation and documentation of forestry decision support tools proposed [1], which are usually followed in reporting on the solutions used. As a forest decision support system (DSS), we understand a software system that can be used for forestry scenario modelling, i.e., for modelling of forest and forest management development over long time horizons based on biological processes and management effects [12]. In any case, the development of future forests depends much on how forest owners and managers act. Bearing in mind that modern forest management needs to cover diversity of economic, ecological, and social aspects, making a decision implies trade-offs, and needs to be implemented through assessments based on computer-driven modelling [13]. Currently available DSSs usually cover the interests of stakeholders with very diverse interests and responsibilities in forest management, ranging from the public or individual forest owners up to national forest agencies [9]. Usually, the forest DSS tools originate from research initiatives which have evolved later into multitask solutions [9]. Quite common features are that the DSSs are best adopted to solve very specific types of forest problems, differing in temporal and spatial scale, spatial context, number of decision-makers, number of objectives, and scope [11]. To deal with new functionality challenges, available DSSs are usually modified, or expert knowledge is applied to extract the facts required based on modelled proxy indicators [12]. In general, the introduction of new forest DSS functionality is usually based on the modernization of already existing solutions. This is normal procedure in countries with well-established traditions in the use of DSSs in forestry. As there are no universally standardized sets of procedures, the development of new decision support tools or the essential modernization of outdated ones could begin with general principles and best practices in user-centred research [14]. This involves prioritizing the preferences and objectives of the users throughout the entire process and employing tools such as contextual inquiry and iterative design. Conducting in-depth observations is typically the most effective approach to gaining insights into users' tasks, workflows, challenges, and, consequently, their needs.

Forestry scenario modelling in Lithuania relates to individual cases of research, mainly based on international projects or computer programs intended for international applications [15–19]. In Lithuania, the most widely used forestry scenario modelling system is *Kupolis* [17]; however, its main drawback is that it is primarily designed for modelling the development of forests and forestry with the minimum changes to the current provisions of forestry models. *Kupolis* has been adapted for use under different climate change and forestry model conditions [15], but it still has remained a system designed and based on the technology of the last century, which does not meet today's realities. The latter assumption is confirmed by the fact that *Kupolis* is used only by individual users and only for the purposes of highly specific research projects. As a result, the need to develop a new forestry scenario modelling and forestry decision-making support system based on the modern

information technology and the needs of current users has arisen in Lithuania. In order to determine the requirements of such a system, it is necessary to identify the potential users and their needs.

Therefore, the main objective of this paper is to explore the needs of the potential users of the forestry scenario system in Lithuania. Then, we also aim to compare the needs specified by different stakeholders with the internationally accepted standards for modern forest DSSs. We acknowledge that there has been no significant improvement in forestry decision support tools in Lithuania since their emergence, and both DSS developers and users have limited experience in this field. Therefore, our objective is also to address methodological issues related to specifying the requirements of modernized forestry DSSs, considering the legal, political, and technological forestry frameworks. Given the limited operational knowledge of forestry DSSs and the potential challenges in exploring a visionary system with unclear specifications, we chose to employ qualitative research methods, specifically conducting in-depth interviews with a select group of well-informed stakeholders. Through this approach, we aim to emphasize the importance of gathering insights from supposed end users to specify a system that has not yet been operationalized.

## 2. Methodology of Research

### 2.1. Study Area

The study was conducted in Lithuania. The total land area of Lithuania is 65,200 km$^2$. Geographically, even though Lithuania is situated in central Europe with central coordinates of 55°10' N, 23°39' E, it has strong historical links with Eastern Europe. Land use development in Lithuania in recent decades strongly depended on the radical societal transformations after Lithuania broke away from the Soviet Union in 1990 and later joined the European Union in 2004 [15]. Lithuania lies on the Eastern European Plain, with characteristic lowlands and hills (the highest point in the country is only 293 m above sea level). In 2023, more than 50% of its land area is used for agricultural purposes. The terrain features numerous lakes and wetlands, and a forest land area covers over 33% of the country [20]. The state's policy is to achieve the forest land proportion of 35% by 2030. Throughout the history of Lithuania, forests have consistently held a crucial position in the country's economy. This significance stems from the fact that wood, being one of the limited domestically accessible raw materials, plays a fundamental role. The importance of other forest functions, such as biodiversity conservation, carbon sequestration, and recreational use, has always been significant. The forest areas in Lithuanian have well-defined property rights. Roughly 57% of the forest lands in the country are owned by the state in the form of national forests and forest reserves, while private owners hold a little more than 42%. Lithuanian forests belong to the European hemi-boreal mixed broadleaved–coniferous forest type in the transitional zone between the boreal coniferous and the nemoral broadleaved forests [21].

### 2.2. Data Collection and Analysis

In-depth interview was chosen as a method of qualitative research to find out the needs of the scenario modelling. There were seven methodological stages used in the study.

1. Definition of the objectives. The purpose of the interviews was closely linked to the overall aim of the study. Firstly, our objective was to identify key forestry stakeholders who are interested in enhancing their management decisions using modern decision support tools. It was important to specify the relevance of each stakeholder's role in forest management. Additionally, we needed to identify the types of decisions for which the stakeholder is responsible, including their approaches to dealing with alternatives. Subsequently, we asked the stakeholders about their expectations regarding specialized tools that would support their decision-making processes. Lastly, we sought feedback from each stakeholder on the interview process and asked for their input on identifying other potential informants.

2. Choosing the informants. Informants potentially interested in the system of forestry scenario modelling and optimisation of forestry decisions were surveyed. In the interviews, a combination of targeted informant selection types (mixed targeted selection) was used:

- For informative elements—the informants who could provide the largest amount of information were selected. Priority was given to informants who are directly responsible for information management or decision support.
- For politically important cases—the informants who were significant in a certain sociopolitical situation were selected, with the objective to analyse a specific case or issue. The most influential actors involved in the development and implementation of forest policy, management of state and private forests, and those actively engaged in forestry education and research, were identified and listed.
- The "snowball" technique is used when the size of the targeted population is unknown and the subjects are difficult to access. During the interviews, each informant was asked to suggest additional candidates for interviews, both from their own institution and other institutions. We then checked for any repeating names and made decisions about whether to interview these additional individuals. Typically, the additional names mentioned by informants overlapped, forming a relatively closed circle of candidates, even though they represented different institutions.
- Theoretical—when the decision regarding which individuals should be included in the study is made according to the first few cases analysed. The interviews began with the actors from the State Forest Service and the State Forest Enterprise, as they have the most experience in working with forestry data and some experience in dealing with forestry scenarios.

Twenty-one informants representing 11 institutions, identified in this paper by reference to their names at the time of the interview, were surveyed (Table 1). Six of them represented the State Forest Service (Lith. Valstybinė miškų tarnyba—VMT), four represented various divisions of Ministry of Environment (Lith. Aplinkos ministerija—ME), two—the State Service for Protected Areas (Lith. Valstybinė saugomų teritorijų tarnyba—VSTT), three—the State company State Forest Enterprise (Lith. VĮ Valstybinių miškų urėdija—SFE), one—the Forest and Land Owners Association of Lithuania (Lith. Lietuvos miško ir žemės savininkų asociacija—LMSA), four informants represented two universities and one research institute, hereafter referred to as research and education institutions (R&E), and one informant was working for a private company engaged in forestry activities. There were multiple informants from some institutions, typically representing different fields of activity and areas of expertise within the same institution. We made sure to avoid overlapping responsibilities within the same institution when selecting the informants. The informants had solid professional experience: it exceeded 20 years for 11 of them and ranged from 15 to 20 years for the other ones (except one informant with 5–10 years of experience). The professional experience of one informant reached 50 years. Practically all informants have a higher education in forestry, 10 of them are Doctors of Sciences. A total of 14 informants have been involved in or participated in scientific activities in some way, while 9 of them have some experience in work related to the scenario modelling, including the use of the European Forestry Dynamics Model (EFDM) and the simulator *Kupolis*.

**Table 1.** Description of the forestry decision support and optimization system informants.

| No. | Organisation | Professional Experience | Functions Performed in the Represented Institution | Education | Participation in Scientific Activity | Scenario Modelling Experience |
|-----|--------------|------------------------|---------------------------------------------------|-----------|-------------------------------------|------------------------------|
| 1 | State forest service | Between 15 and 20 years | Head of a subdivision | Forestry | | |
| 2 | State forest service | Between 15 and 20 years | Deputy head of a subdivision | Forestry | | |

**Table 1.** *Cont.*

| No. | Organisation | Professional Experience | Functions Performed in the Represented Institution | Education | Participation in Scientific Activity | Scenario Modelling Experience |
|---|---|---|---|---|---|---|
| 3 | State forest service | Between 15 and 20 years | Head of a subdivision | Forestry, Dr. | Yes | |
| 4 | State forest service | Over 20 years | Head of a subdivision | Forestry | Yes | Yes |
| 5 | State forest service | Over 20 years | Sen. specialist | Forestry, Hb. Dr. | Yes | |
| 6 | State forest enterprise | Over 20 years | Deputy manager | Forestry | Yes | Yes |
| 7 | State forest enterprise | Between 15 and 20 years | Specialist | Forestry | Yes | Yes |
| 8 | State forest enterprise | Over 20 years | Deputy manager | Forestry, Dr. | Yes | Yes |
| 9 | Forest and Land Owners Association of Lithuania | Over 20 years | Manager | Forestry, Dr. | Yes | |
| 10 | Ministry of environment | Between 15 and 20 years | Manager | Forestry, Dr. | Yes | Yes |
| 11 | Ministry of environment | Over 20 years | Head of a subdivision | Forestry | | |
| 12 | State Service for Protected Areas | Over 20 years | Manager | Forestry | | |
| 13 | State Service for Protected Areas | 5–10 years | Specialist | Biology | | |
| 14 | Ministry of environment | Between 15 and 20 years | Head of a subdivision | Forestry | Yes | |
| 15 | Ministry of environment | Between 15 and 20 years | Sen. specialist | Ecology and public administration | | Yes |
| 16 | Research and education | Between 15 and 20 years | Manager | Wood science, Dr. | Yes | |
| 17 | Research and education | Over 20 years | Sen. specialist | Forestry, Dr. | Yes | Yes |
| 18 | State forest service | Between 15 and 20 years | Deputy manager | Forestry, Dr. | Yes | |
| 19 | Research and education | Over 20 years | Manager | Forestry, Dr. | Yes | Yes |
| 20 | Research and education | Over 20 years | Sen. specialist | Forestry, Dr. | Yes | Yes |
| 21 | Private forestry company | Over 20 years | Manager | Forestry | | |

3. Development of the interview protocol. For the purposes of the surveys, questionnaires with open-ended questions to elicit detailed responses from participants were drawn up. The principle on which the questionnaires were based was to determine the area of work and interest of the informant first and to follow with the discussion of the informant's need of information related to forestry and the use of such information for the purposes of his/her job functions. After that, the decisions to be made by the informant are discussed,

trying to relate these decisions to the use of relevant information. This is followed by questions about the need for information on future forest conditions for decision-making purposes, about the necessity to consider the alternatives, the need for optimisation, and about the need for a decision-making support tool. At the end of the survey, the informants are asked to identify other experts whose opinion could be important. The questionnaire is provided in Annex 1, which can be found at the end of this paper.

4. Training the interview team. The interview teams gained experience by conducting dummy interviews with other members of the research team who were not involved in the study. The questionnaires were adjusted based on the outputs from these training sessions.

5. Conducting the interviews. Only face-to-face interviews were conducted. The survey was carried out by two persons: one of them was directly communicating with the informant, and the other was taking notes and made an audio record of the interview. Note-taking during the interview also involved capturing nonverbal cues and contextual information. The audio record was made subject to the informant's consent. The interview lasted from 30 to 140 min, and the average duration was about one hour.

6. Recording and transcribing. After the interview, the audio record was examined and a transcript of the interview was drawn up.

7. Analysing the data. The survey data, summarised responses of the informants, were organised in MS Excel tables, where they were grouped according to the informants and questions. Each interview was analysed using qualitative analysis methods: a search of repetitions and contradictions, identification, and refinement of ideas relevant to the study was performed, and the results obtained were summarised and described. Quotes were used to preserve originality and to reproduce the thoughts of the informants as accurately as possible. They are provided in Appendix A, which can be found at the end of this paper. To avoid repetition, to clarify the informant's idea on a specific issue and to convey the message in a shorter way, some quotes were shortened by cutting out less relevant content, marked by (...), and inserting certain phrases in parentheses instead of the words missed by the informant based on the context of the entire question, conveying the message of the informant more clearly to the reader. The informants were encoded by giving a number according to the date and time of their survey (Table 1). More information about the informant can be obtained from the authors of the current paper. The names of the informants are known. Citations given in this paper are translated from Lithuanian into English. One interview was conducted directly in English.

## 3. Results

The needs of the forestry scenario modelling system in Lithuania, extracted through qualitative in-depth interviews, are summarized below. These needs indicate the main functional and nonfunctional requirements for the system, as well as specific attitudes of different actors involved in forestry decision making at various levels. For a more detailed introduction to our findings, including quotes from the interviews associated with specific informants, please refer to Annex 1. The vast majority of informants agreed that a system of forestry scenario modelling and optimization of forestry decisions is important and necessary. The purpose of this system is to support decision making in forestry while providing a scientific basis for such decisions. Its role extends beyond assisting in decision-making decisions and evaluating alternatives; it also aims to convince senior decision makers, support proposals, and enhance their informativeness and comprehensibility for the professional societies and the general public.

Although most of the informants traditionally show interest in using highly detailed information of forest resources, such as at the level of individual forest compartment or even single trees, there is recognition of the importance of working at other scales, such as the country level or in a region. The desired characteristics to be modelled align with the data collected during the inventory. Informants expect more than just modelling of wood and stand characteristics. Alongside volume and other dendrometric attributes, there is an emphasis on economic evaluation, specifically assessing the economic benefits of different alternatives or

solutions. The evaluation of other forest ecosystem services, including conservation, recreation, carbon capture, and potentially the social significance of the forest, is also considered important. In this context, informants assign the role of the forest ecosystem inventory to the scenario modelling subsystem. Consequently, the results of the modelling are expected to provide information that enables the quantification of sustainable forestry based on the evolving economic, ecological, and social functions of the forest. The forestry scenario modelling system should not only provide the quantitative estimates of forest ecosystem services, but also discuss potential risks associated with various forestry decisions.

The functional requirements of the scenario modelling system encompass the need for tools that facilitate forestry planning and enable the assessment of various alternatives based on economic, ecological, and social justifications. The system should evaluate the impact of forestry measures over different periods, including immediate effects and throughout the entire forest stand rotation period. Additionally, it is recognized that forestry should be evaluated with respect to the priority forest ecosystem services, such as economic value, biological factors, social aspects, protection, carbon capture, and more. Recording the effects of climate change and incorporating them into the modelling procedures is a crucial requirement for the forestry scenario modelling system. This entails describing not only changes in forest growth under different climatic conditions but also the ability to utilize different forestry programs both under the normal and extreme conditions (e.g., storms, pests outbreaks, etc.). Lastly, it would be desirable for the modelling system to allow differentiation between state and private forests.

The system should possess an openness to further development and enable the modelling nonstandard forestry options that are not covered in current forestry textbooks or existing legislation. It must be adaptable to the development of national forest programs, planning forest expansion, and fulfilling the country's commitments to combat climate change. Additionally, the forestry scenario modelling system should incorporate functionality for estimating the costs associated with different alternatives, including calculating compensations to owners for imposed restrictions.

If the forestry scenario modelling system is implemented as a computer program, it should allow the user the ability to customize the system by integrating new modelling tools, among other features. Ideally, it should be accessible not only to specialists who prepare the decision proposals but also to decision-makers themselves. The involvement of the scientific community in the design, development, and utilization of the forestry scenario modelling system is also vital. It is also important to establish a clear vision for its development and modernization, while recognizing that maintaining and supporting the system will entail costs.

The system should aim to encompass not only cover forest processes but also the evaluation of the current or potential changes in land use. Emphasizing the significance of assessing forest changes, the system should leverage retrospective information on both forest resources and land use. Furthermore, the forestry scenario modelling system should not only strive to incorporate data on forest resources but also consider the broader interests of the forest sector or the national economy. The tool is expected to provide the most probable scenario, along with its probability, as well as alternative scenarios.

Table 2 presents a summary of the requirements for the forestry scenario modelling system. The assessment criteria were chosen using the engineering method, which involved a thorough examination of the detailed responses provided by the informants. The most demanding informants for the forestry scenario modelling system were scientists from research and education institutions. They provided positive responses to all groups of questions. None of the informants, except the scientists, had any specific requirements regarding the nonfunctional aspects of the scenario modelling system, such as user interface and customization options. Experts from the State Forest Service and the State Forest Enterprise, responsible for forest management planning, showed interest in all aspects of the system except for nonfunctional requirements. It should be noted that they are directly responsible for generating or supervising forest management decisions, although not for operational implementation.

Practical foresters from the State Forest Enterprise did not express opinions beyond their direct professional functions or competencies, such as forestry risk and climate impact assessments, testing alternatives, or entering the domain of private forestry.

**Table 2.** Summary of forestry scenario modelling system needs.

| Various Aspects of the Scenario Modelling System | VMT * | SFE1 | SFE2 | ME1 | ME2 | VSTT | LMSA | R&E | Private Company |
|---|---|---|---|---|---|---|---|---|---|
| The need for forestry scenario modelling and decision optimisation system: | | | | | | | | | |
| To address national challenges | + | + | + | + | ? | + | ? | + | - |
| To address estate or stand-related challenges | + | + | + | - | ? | + | + | + | + |
| The desired level of detail for future forest resource information: | | | | | | | | | |
| Aggregated country-level information | + | + | + | + | | | | + | |
| Stand-level information | + | + | + | + | + | + | + | + | + |
| The desired contents of information on future forest resources: | | | | | | | | | |
| Dendrometric characteristics of forest resources | + | + | + | + | | + | + | + | + |
| Economic evaluation of forestry | + | + | + | + | | | | + | + |
| Assessment of other ecosystem services | + | +? | + | - | + | + | | + | |
| Assessment of forestry risks | + | | + | | | + | | + | |
| Requirements for the functionality of the scenario modelling system: | | | | | | | | | |
| Modelling of forest growth | + | + | + | + | | | + | + | + |
| Modelling of forestry measures | + | + | + | + | | + | | + | + |
| Evaluation of forestry | + | + | + | + | | | | + | + |
| Evaluation of alternatives | + | | + | ? | + | + | + | + | + |
| Climate change impact assessment | + | | + | + | | | + | + | |
| Assessment of the impact of forest ownership | + | | + | + | | | | + | |
| Possibility to implement nonstandard forestry models | | | | | | | | + | |
| Nonfunctional requirements for the scenario modelling system: | | | | | | | | | |
| Possibility of customisation | | | | | | | + | + | |
| User interface accessible to the decision maker | | | | | | | | + | |
| Vision of development and modernisation | | | | | | | | + | |
| Interfaces with land-use scenario modelling: | | | | | | | | | |
| Functionality for modelling the land use development | +? | -? | + | + | + | + | | + | ? |

+ At least one informant from the respective institution gave a positive comment about the significance of a particular aspect. - An informant from the respective institution gave a negative comment about the significance of a particular aspect. ? The opinion of the informants from the respective institution was unclear. If no marks are available, the informants from the respective institution did not mention the specific aspect during the interview. * Abbreviations used in the table: VMT—the State Forest Service (Lith. *Valstybinė miškų tarnyba*); SFE—State enterprise State Forest Enterprise (Lith. *VĮ Valstybinių miškų urėdija*). SFE1 denotes the experts of SFE, responsible for forestry, while SFE2—the experts, responsible for forest management planning; ME—Ministry of Environment (Lith. *Aplinkos ministerija*). ME1 denotes the division of the ME coordinating forestry issues, while ME2—the division responsible for protected areas and landscape management; VSTT—the State Service for Protected Areas (Lith. *Valstybinė saugomų teritorijų tarnyba*); LMSA—Forest and Land Owners Association of Lithuania; R&E—research and education.

Representatives from the Ministry of Environment did not require functionality related to working with forestry scenarios at the stand or estate scales. This is quite evident, as the institution responsible for building forest policy in the country is not accountable for local-level forest management. There were also notable differences in needs between the two groups of Ministry of Environment experts. The interests of informants responsible for protected areas and land management planning were less demanding, as their involvement in forest management was minimal. Their demands were usually minor and focused on general aspects of forestry. It is not surprising that they were interested in functionality related to assessing ecosystem services other than timber supply and modelling land use development.

Informants from the State Service for Protected Areas, despite being under the Ministry of Environment, expressed greater interest in forestry scenario modelling compared to their superior institution. They are directly responsible for implementing state policies and strategies in the field of protection and management of protected areas, which also includes some forest land. Although the State Service for Protected Areas is not directly responsible for operational forest management, they are still an important stakeholder in this field.

Representatives from the Forest and Land Owners Association and private forest enterprise had similar expectations, primarily related to local-level forest management decisions, usually at the estate and forest stand levels. It was evident that the focus of representatives from private forest companies was on timber-supply-oriented forestry needs at the estate level.

## 4. Discussion and Conclusions

The qualitative research method, based on in-depth interviews with a limited number of relevant informants, was employed to explore the needs for a forestry scenario modelling system among Lithuanian forestry actors. This approach is commonly used in social sciences [22–24]. In the field of forest decision support systems, conducting in-depth interviews with relevant stakeholders before constructing the system can provide valuable insights into user needs. Therefore, through this study, we also demonstrate the use of qualitative research for conceptualizing information systems, along with a discussion of the pros and cons of this approach. The qualitative research method, as an alternative to a quantitative survey based on questionnaires, was chosen for the following reasons:

- Qualitative methods are typically preferred when there is a need for a thorough exploration of stakeholders' perspectives, experiences, and requirements. Qualitative research offers a deeper insight into the issue at hand, particularly when it is not well understood. This has been the case in Lithuanian forestry, which has struggled with command-and-control forest governance, excessive regulation and control, low adaptiveness, and prioritization of opinions from "mighty" experts [25]. Gathering valuable inputs from stakeholders with limited understanding of the role of forest decision support systems through quantitative surveys would be extremely difficult. Qualitative research facilitates the assessment of various indicators that are difficult to express in clear, measurable terms, such as the wishes, visions, or intentions of users. Furthermore, the outcomes of improved forest management decisions facilitated by decision support systems (DSSs) are often intangible and challenging to quantify [26]. While this survey did consider current processes in forest management decision-making, the focus was placed on not-yet-available approaches.
- The fundamental weakness of the quantitative research method, as an alternative to the qualitative approach, is associated with the preparation of a survey questionnaire, its comprehensiveness, and objectivity [27–30]. Researchers may be insufficiently familiar with the subject of the study and overlook important aspects or inadvertently influence the conclusions by emphasizing certain aspects in the questionnaire.

Qualitative research does come with certain limitations, including the collection of large volumes of data (processing audio recordings and preparing verbatim reports being the most time-consuming stage), the risk of subjectivity in data interpretation, and limited possibilities for generalization of observations. Therefore, careful planning is required when

selecting the number of informants and their inclusion [22]. Studies based on the wishes, visions, and intentions of the informants may even be considered as extending beyond science. However, due to the use of qualitative methods, our study can be considered truly inductive research, without the mandatory need for predetermined hypotheses.

Qualitative methods are commonly used in social sciences and have also been applied in forestry research, including in Lithuania. For instance, in-depth interviews and qualitative analyses were employed to provide detailed contextualized portrayals of private forest owners in Lithuania [27]. Additionally, these methods were utilized to characterize the principles underlying forest governance, such as the preference for the opinions of established experts over alternatives outside the current legal framework [25]. In another study, a qualitative approach was used to identify different types of forest owners and specify a set of forest management programs, which facilitated the simulation of forest owner behaviour for modelling various forestry scenarios in the DSS *Kupolis* [15]. Nonetheless, the utilization of qualitative methods in studying the potential of forestry scenario modelling and forest decision support systems remains relatively limited. There are numerous publications on existing solutions, including those based on reports from DSS users and scientific publications [2,10,11,31,32]. Some studies have included in-depth interviews with system developers and advanced users [12,33]. However, these mostly represent the perspectives of system developers or advanced users. The role of forestry stakeholders is often linked to their involvement in the decision-making process, such as utilizing multiple criteria decision support methods in participatory forest management, where understanding user needs through qualitative methods like in-depth interviews is crucial for developing or applying effective decision support systems [11,34–41]. Qualitative methods can also be efficient in guiding strategic decisions, especially when only qualitative information is available [42]. Nonetheless, it is widely acknowledged that three factors contributing to the overall success of a DSS should be considered during its development: (i) characteristics of the DSS, (ii) characteristics of the users, and (iii) the process of introducing the DSS to its users [33]. Therefore, it is crucial to couple the models, data, analytical engines, processes, and user interface of the DSS—referred to as DSS research—with a comprehensive exploration of DSS users and their needs, to effectively operate a modern DSS.

The current study is based on data gathered from a relatively small sample of forest DSS stakeholders in Lithuania. This approach limits the statistical inference to other populations, such as private forest owners. However, it is widely accepted that the focus in qualitative research is not on the sample size but on collecting detailed information about the individuals under analysis [22]. For case study research, it is even recommended to interview no more than 4–5 informants [22]. Beyond this number, the responses may become redundant, offering little new insight. The study included a limited number of institutions. This selection is because there are only a few institutions in the country that have the capacity to effectively implement their interests in forest management and the use of DSSs [43]. While there were several informants from the same institution, we ensured that they represented different competencies and responsibilities, avoiding the development of a collective narrative.

There are examples of using quantitative questionnaires in studies on increasing the usefulness (i.e., the range of tools incorporated in the DSS) and perceived usefulness (i.e., the impact of the DSS on job performance) of forestry decision support systems [44–47]. However, these studies have primarily focused on the post-implementation evaluation of information systems. Nonetheless, these studies have encountered issues related to a limited number of respondents, which restricts the statistical significance of highly specific questionnaires.

To summarize the methodological approach for studying the needs of DSS users in Lithuania, we conclude that implementing qualitative questionnaires or desktop research would be highly challenging due to the following reasons:

- There is no operational DSS in the country, meaning it is primarily used within research projects. As a result, it is extremely difficult to provide objective judgments and unambiguous proposals.

- Competence levels regarding the requirements for forestry DSSs are still at the visionary stage. Therefore, stakeholders are more comfortable elaborating on these matters during interviews.
- The number of productive informants would inevitably be limited, which would restrict the ability to draw statistically sound conclusions due to the scarcity of influential and well-informed stakeholders.
- There are examples of successful application of qualitative methods in the country for other tasks involving the same stakeholders.

It is important to revisit the methods used to specify user needs as well as the needs themselves as forestry DSS development progresses.

It was not surprising that the majority of surveyed forestry stakeholders in Lithuania identified a system for modelling forestry-related scenarios and optimizing management solutions as important and necessary. The purpose of such a system would be to enhance support for the decisions made by experts and presented to decision-makers, providing them with a stronger scientific foundation and enabling the assessment of alternatives using objective methods. The use of forestry scenario modelling in decision making was expected to result in more informative and comprehensible decisions for both professionals and the general public.

The specific requests and proposals regarding the forestry scenario modelling system were strongly influenced by the functions performed by the informants and their understanding of the concept of scenario modelling. While experts in forestry scenario modelling agree that the system should operate at the country level, they are still interested in utilizing highly detailed forest resource information, particularly related to stand-wise forest inventory, for the development of internal forest management projects. It is believed that the forestry scenario modelling system should incorporate tools for specifying various silvicultural programs, evaluating alternatives, and providing support from economic, environmental, and social perspectives. Furthermore, the forestry scenario modelling system should be open for further development and accessible not only to a limited group of experts but also to decision-makers and the scientific community.

Nevertheless, the vision for advanced forest DSSs in Lithuanian forestry does not deviate significantly from modern international standards. The requirements and characteristics of modern forest DSSs can be derived from scientific sources [9,10,48–50]. Below, we list these properties and evaluate the perspectives of the informants in relation to them.

Firstly, a modern forest DSS needs to encompass the full range of forest ecosystem services. This means that it should not only project timber and biomass production, which was agreed upon by all informants, but also consider options for biodiversity conservation, carbon sequestration, and other regulatory services, as well as recreational and aesthetic values. The majority of informants supported the inclusion of nontimber forest products and services. Most informants also emphasized the importance of the DSS's ability to incorporate changing market prices for all forest products and services.

Regarding the inclusion of climate change effects, there were differing opinions among the informants. However, they unanimously insisted on the inclusion of climate change impact assessments. The requirement for spatial specificity in landscape-scale analyses, specifically the use of information on the location and spatial relationships of forest stands, was omitted from the forest DSS requirements. Spatial planning components in current forest management planning are typically associated with human efforts, albeit with computer assistance. Nevertheless, the integration of land use change modelling was considered an important functionality of forest DSSs.

The inclusion of forest owner and manager behaviour was largely neglected by the informants, primarily due to existing traditions and legal requirements that apply uniformly to all types of forest owners and managers [25]. However, it should be noted that special attention should be given to forest owner preferences and management peculiarities, as evidenced by forestry scenario analyses showing significant differences in outcomes under the same legal forestry framework [15].

While most informants recognized the importance of evaluating alternatives, these alternatives were typically considered within the confines of the current forestry paradigm. The possibility of discussing alternative forest management options that extend beyond the legal forestry framework was only accepted by researchers. This indicates that the majority of key forestry actors remain sceptical about adopting new forestry approaches, even if they are virtually exercised within forest DSSs.

The informants typically requested features for the proposed forestry scenario modelling system that aligned with their current roles and responsibilities. This can be attributed to the dominance of strict regulations, which still prevail over decision-making freedom for forest owners and managers in various forestry aspects within the country [25]. Tasks assigned to forest managers are generally well documented, leaving little room for elaborating alternative decisions. Furthermore, the forest policy arena is heavily influenced by state forest institutions [50]. As a result, the overall success of progress in forestry scenario modelling relies heavily on the decisions made by key state-controlled actors. Also, it is a common practice across various industries to initially implement new information technology by focusing on conventional tasks, followed by continuous improvement and expansion [51]. Therefore, a suitable strategy for implementing forestry scenario modelling solutions in Lithuanian forestry could involve addressing the current needs and required functionalities within the forest management planning branch of the State Company State Forest Enterprise and the State Forest Service. This development, implementation, and validation can be complemented by support from research and education institutions. In principle, our findings regarding the needs and expectations of users of modernized forestry DSSs assume indirect validation of methodological questions through the specification of requirements using qualitative interviews. However, the results of this study have been utilized in planning real-life DSS implementations, thus confirming the methodological soundness of the approach.

In conclusion, this study highlights the diverse perspectives, wishes, visions, and intentions of key Lithuanian forestry actors regarding the aims, objectives, and essential functionality of forestry scenario modelling tools. It is evident that the understanding of the requirements for modern forest decision support systems (DSSs) is greatly influenced by the current forestry paradigms in the country and the professional experiences of individual informants. Nonetheless, the expected functionality of the planned forest DSSs aligns with modern international standards.

We found that the utilization of qualitative research through in-depth interviews is an effective approach to delineate the specifications of a modern forest DSS. This specifically refers to exploring the users' needs for a system that has not yet been operationalized. It helps mitigate preconceptions and address gaps in the vision of the desired product. The findings of this study were utilized to develop a framework of core solutions for the forestry and land-use scenario modelling subsystem of the Lithuanian National Forest Inventory Information System. The insights gained from this study can serve as valuable lessons for developing a set of procedures for specifying new DSSs and modernizing existing ones in the field of forestry.

**Author Contributions:** Conceptualisation, G.M. and D.J.; methodology, G.M. and D.J.; validation, G.M.; formal analysis, D.J. and G.M.; writing—original draft preparation, G.M. and M.P.; writing—review and editing, D.J. and J.V. All authors have read and agreed to the published version of the manuscript.

**Funding:** This research paper has received funding from Horizon Europe Framework Programme (HORIZON), call Teaming for Excellence (HORIZON-WIDERA-2022-ACCESS-01-two-stage)—Creation of the centre of excellence in smart forestry "Forest 4.0" No. 101059985. This research has been co-funded by the European Union under the project "FOREST 4.0—Center of Excellence for the development of a sustainable forest bioeconomy", No. 10-042-P-0002.

**Institutional Review Board Statement:** Not applicable.

**Informed Consent Statement:** Not applicable.

**Data Availability Statement:** Data available on request.

**Conflicts of Interest:** The authors declare no conflicts of interest.

## Appendix A. A Summary of the Responses Provided by the Informants

| Questions | Summary of the Responses from the Informants and Selected Citations (Hereafter, the Numbers in Parentheses Refer to the Corresponding Informants' Numbers in Table 1) |
|---|---|
| **Part 1. The area of work/interests of the informant** | |
| 1.1 The organization/institution you work for | State Forest Service—6, Ministry of Environment—4; State Service for Protected Areas—2; State company State Forest Enterprise—3; Forest and Land Owners Association of Lithuania—1; Research and education—4; Private company engaged in forestry activities—1. |
| 1.2 What is your professional experience? | From 5 to 10 years—1; From 15 to 20 years—9; Over 20 years—11. |
| 1.3 Your duties and key functions. | Head or deputy of subdivision—6; Manager or deputy manager—9; Specialist or senior specialist—6. |
| 1.4 Education | Forestry 18; Ecology and Public Administration—1; Biology—1; Wood science—1. |
| 1.5 Do you participate in any scientific activities? | Yes—14; No—7. |
| 1.6 Have you ever faced the modelling of scenarios for development, the application of decision-making support systems or three-dimensional modelling? | Yes—8; No—13. |
| **Part 2. The use and needs of forest information** | |
| 2.1 Do you need to use information about forests in your work? 2.1.1 What forest information do you use (please specify the level of detail and the sources of information)? 2.1.2 What institutions or organisations do you cooperate with on forest information issues? | 2.1.1 All informants acknowledged that information about forests is necessary or even indispensable in their everyday professional activities. Although the level of detail and purpose of the sources used are different, most of the informants pointed out that they used the data of forest inventory, forest statistics and state cadastre of forests: *"We use all databases which have any connection with the forest"* (6); *"(...) In particular, [we need] general information from the statistical yearbook of forestry based on national and stand-level forest inventory. One [source is used] for strategic planning, while the other one is used for silvicultural treatment planning in forest enterprises for"* (8); *"[We use] almost everything about forests what is publicly available online—from statistics to the cadastre"* (14); *"Yes, most often 2 sources [are used]: standwise forest inventory which covers the whole of Lithuania, and if you need something more detailed, there is VMT database, we have the access. Another [source] is NFI, but it is difficult to access, there is no direct access"* (19); *"[We use] information database of forests: characteristics of forest stands, information about the accomplished silvicultural treatment [and other information] starting from Google maps finishing with VMT databases, all accessible information on silvicultural treatment activities"* (21). 2.1.2 The informants are in contact with a very large number of Lithuanian and international institutions on forest information issues: *"State Forest Enterprise, the Ministry of Environment, the State Tax Inspectorate, forest enterprises, municipalities, FAO, Eurostat, the Centre of Registers, forest owners, forest managers, forest planners, NMA [National paying agency], the Ministry of Economy, the Ministry of Agriculture, the National Land Service"* (2). The nature of the communication depends on the functions of the informant's institution and interests; some of them were developers and providers of such information: *"We receive information all the time, [it] is updated using forestry cadastre data. We collect it from the forest enterprises what they change it. We get so little in comparison to what we would like to receive from them. There are other sources [used] to learn what happens in the forests and protected areas, everything that is going on is recorded. We provide information to anyone in the country, they get what they want what they desire. [We provide] detailed and statistical information, (...) we provide information to the Department of Statistics, Eurostat and the FAO and the United Nations, their economic commission, there is an organisation European Forests..."* (5). Some informants are users: *"[We get information without having to create it from] VMT and all available sources"* (21). This, of course, can affect the attitude of the informants to the need of information on forest resources and, in turn, to the particularities of manipulation of this information. |

| Questions | Summary of the Responses from the Informants and Selected Citations (Hereafter, the Numbers in Parentheses Refer to the Corresponding Informants' Numbers in Table 1) |
|---|---|
| 2.2 How and for what purposes do you use forest information? | The use of information on forest resources is directly related to the functions of the informant and the institution he/she represents. Among the large number of areas of application of information on forest resources, some areas are directly related to the purpose of NFI (National Forest Inventory), addressing nationwide forestry issues: *"(…) [we use the information] of the national level to support different forestry policy decisions (…), this is related to decisions at national level, (…) for me [this is important, relevant] for the national forest programme"* (10); *"(…) "[We use the information] for policy-making and preparation of legislation and in specific cases when queries are made"* (11);*" [Uses of information:] NFI carried out sampling method (…) GHG [Green-house gas] is a convention on climate change, Kyoto Protocol. Besides, there is forest state monitoring, the country's obligations under the Long Range Trans-boundary Air Pollution"* (4); *"[Uses of information:] "The national inventory for general purposes, (…) in addition, international activities, international commitments, FAO, the United Nations, all the commitments where you have to provide information. All kind of conventions, Kyoto Protocol and similar, what relates to forestry"* (18); *"[We use the information] for management strategy development"* (21). |
| 2.3 What kind of information about forests do you lack (please specify the nature/level of the lacked information)? | When asked what kind of information about forests is lacked, most of the informants spoke not only about the lack of information, but emphasised its reliability, age, and compatibility: *"The information is insufficiently reliable (…), the information is not accurate"* (2); *"Now the information (…) must be fresh, that's what is missing. All planning and efficiency depends on the information we have. (…) changes in the forest, they occur (…) changes are recorded, but there are no channels for us to get the information (…), that information is kept, but we are unable to use it"* (5);*"I do not lack anything, I just want it to be made more specific sooner, separated from other pieces of information, there are large quantities of information, many databases, just everyone describes it differently, I just want it to be organised"* (6); *"What is being developed now—to collect consistent geoinformation on forests: block boundaries, borders, the coordinates of linear objects located inside, in order to have a consistent basis for any operations. We lack information about any changes implemented in forest enterprises, what is harvested, what roads are being repaired, we lack an information system, notification in cyberspace on what was done"* (18); *[We lack information] about performed silvicultural treatment, (…), what was done, about harvesting, the information of VMT is old, in particular information about volumes, if inventory is old"* (21). In the context of this study, the following information is stated as missing: *"There is a lack of information about the value in ecosystem functions which probably [would] be more useful when discussing with politicians and the public. Now everyone gets all those [ecosystem] functions free of charge (…) it would be important to [know the economic value and the value of economic functions]."* (8); *"[There is a lack of] all dynamics of [forest information, characteristics], not only for the change of species"* (2); *"[There is a lack of] archival information on forest land (…)"* (3); *"The information which we provide and which is missing: I emphasise the issue of the future forest development. The Department [Ministry of Environment] assumes obligations regarding the expansion of forest areas and change of other indicators. There [is] a set of basic indicators, and commitments are made regarding their change, if the change is different, one looks why. They [the indicators] are evaluated using the engineering method involving individual people, they consider what commitments can be made. (…) the system would allow making such forecasts more accurately"* (4); *"There is a lack of economic information on forestry (…)"* (17). Thus, already at this stage of the survey, some informants identified the need to expand access to the countrywide forest information and applications, and to focus on the information about the future forest resources. By the way, the informants also identified their specific needs or forest resource information gaps which could be considered in the development of the country's forest information content: *"I think the greatest lack of information is where you could be involved in its generation, aggregation. Sometimes information is good, but it is one-sided, it does not expand the view, a broader view is lacking. In particular, NFI—if you need something, you have to make a specific request. In respect of NFI, there is the main thing—it is not yet available to the user on the basis of GIS. (…) if you want [information] on the map from different angles, it is missing"* (19); *"Reliable information about private forests and what is going on in them: statistics and owner behaviour. There is lack of information about silviculture treatment in private forests, there is lack of economic information in terms of taxes. As concerns forest-[related] direct information at national level, I would not say that something is missing, on the contrary, the abundance leads to the desire to do more and sometimes hinders decision-making. But there is a great lack of [information] about private [forests]. Information about people, economic and social matters, not about the tree"* (10). |

| Questions | Summary of the Responses from the Informants and Selected Citations (Hereafter, the Numbers in Parentheses Refer to the Corresponding Informants' Numbers in Table 1) |
|---|---|
| Part 3. The needs of forestry scenario modelling | |
| 3.1 Do you adopt decisions which require using information about forests? 3.1.1 Please provide a brief description of such decisions and the type of information about forests required for them. How important are the territorial division of information, the characteristics of forests, and the ecosystem services provided by forests? | The need for the forestry scenario modelling system is primarily related to the decision making that determines forestry. The informants included some individuals who directly made decisions themselves, but a significant number of them were persons implementing such decisions or advising the decision-makers: *"We do not really make decisions, our mission is to implement them. (...) We are widely involved in a variety of decisions from uses to forest programmes (...)"* (5); *"it's more [our] suggestions, recommendations"* (16); *"I am a scientist, I do not adopt management decisions, we develop tools and programmes that help to make decisions"* (20). Especially important is the latter observation—the developer's task is to provide a tool in accordance with which the decision-makers would perform their functions, but not to predict all the possible decisions. |
| 3.2 Is the information on forest areas, forest condition, forest ecosystem services and future benefits important for the decisions you make? 3.2.1 What decisions do you make the adoption of which requires information on future forest development? 3.2.2 What characteristics of future forests would you need? | Regardless of their relationship with the decision making, the informants discussed the need for information on forest areas, the state of forests, the services provided by forest ecosystems, and the benefit for any decisions to be adopted in the future. 3.2.1 The informants emphasised the need to know information about forests in the future, because it would reduce the uncertainty of today's forestry and political decisions: *"When planning silvicultural treatment, it would be sound to know what parameters may be in the future. Prediction of stand parameters in the future might be able to affect silvicultural treatment planning"* (1); *"We need to understand that silvicultural treatment already focuses on the future, with the objective to have a good forest in the future. A [forestry] project is for ten years at the minimum, it is made within the framework of rules, but must take account the future vision to the extent permitted by those rules. We always bear in mind that we must always have a picture of the future when planning now"* (6); *"We need to know how much, when, and where the ecosystem services can be expected, not only instantaneously, but also over time, and to evaluate them economically (...)"* (19); *"A thing of interest for me in the future is climate change, and in creating scenarios it is important to anticipate what it would be if, e. g., precipitation increases, or the average annual temperature rises"* (20). When talking about the future, the informants mentioned the significance of preparation of possible future alternatives—scenarios: *"The Department [Ministry of Environment] assumes obligations regarding the expansion of forest areas and change of other indicators; there is a set of basic indicators, and commitments are made regarding their change(...) the system would make those predictions more accurately and even choose several ones, if several scenarios [could] be modelled. Both for the ME and industrialists is important to know what to expect from forest and what to focus on and then optimise—to choose ways. [One] more area related to forests is the greenhouse effect, forest forecasts are very important for that. Looking at the experience of other countries, standwise forest [inventory] may show increasing volume, therefore the selective one is required [for self-checking before making important commitments], although it also has certain shortcomings. Scenario modelling is extremely important for climate change and reports"* (4); *"For us, those scenarios are important in terms of Lithuania's future ability to develop silviculture and to maintain [preserve] species"* (14). 3.2.2 In answering the question of what future forest characteristics they would need, the informants tended to emphasise the dendrometric characteristics of forests, although some of them noted the need for ecosystem services or economic values: *"Of course, I would need future dendrometric characteristics: height, volume, density..., this is what we strive for when planning"* (6);*"Recreational assessment [is required], [future] stand parameters"* (2);*"Area characteristics [are required]: how much forest land there [will be] and [how it will] change. This is related to afforestation and deforestation, these are the main characteristics looking to the future. In addition, [it is related to] reforestation: the scope of planting and its significance. Further, there are the characteristics of volume: the volumes, how they are going to change, volumes by species, assortment structure and how it will change. Increment: wood volume change, yearly harvesting, tree death and utilisation of the dead part. Now, especially due to climate change, tree death, winds, their forecasts, "what ifs" with the winds changing and the resulting changes of the characteristics [are important]. [The future] characteristics of damage [are also necessary]: defoliation and other damage and, their development, changes in accumulated deadwood in the forest, composition of species"* (4); |

| Questions | Summary of the Responses from the Informants and Selected Citations (Hereafter, the Numbers in Parentheses Refer to the Corresponding Informants' Numbers in Table 1) |
|---|---|
| 3.2 Is the information on forest areas, forest condition, forest ecosystem services and future benefits important for the decisions you make?<br><br>3.2.1 What decisions do you make the adoption of which requires information on future forest development?<br><br>3.2.2 What characteristics of future forests would you need? | *[We would be interested in the future] stand age, maturity [age structure and quantity of mature trees], areas. It could be the naturalness of [future] forests, whether planted or natural"* (13); *"[We need] the volume, composition of species, increments [of future stands], what will be in the forest after a certain time and what its value will be"* (21); *"I think, starting with the amount of carbon stored in the forest, wood quality characteristics, species of trees that can live and the impact of climate change on them, on the increment and how it would determine the future decisions"* (19). Although the desired future forest characteristics seem to be quite chaotic at the first glance, they are subject to the principal requirement to replicate the content of data collected in current inventories. Some informants mentioned the specific future information which is directly related to their work: *"Climate change. There are talks already that it is better not to plant fir groves. Or maybe it will be better not to plant them only after 100 years. Their future use, the main characteristics, you cannot catch all the details. The closer in terms of time, the finer economic and social issues become important in the near future and [they need to be linked] to industry, population, demographic matters"* (16); *"The forestry data which would allow to predict the existence of a habitat, dead wood, herbaceous vegetation, more botanical [data about] the spatial structure of the stand. And their changes"* (14); *"We do not know the ecosystem services, the quantities of carbon stored and the oxygen emitted, the extent to which water runoff is suspended, another thing is the social factor"* (15). Here, we can also see a desire for information that would allow to judge on the sustainability of future forestry, i.e., the coherence of the economic, ecological, and social functions of the forest. |
| 3.3 To what extent and how are the decisions you make related to the evaluation of different alternatives? In other words, how often do you find yourself asking the question 'what if?' | A portion of the informants answered the questions "To what extent and how the decisions you make are related to the evaluation of different alternatives?" and "To what extent and how the decisions adopted by you could be changed by a scientifically based tool that enables optimisation of the decisions adopted which performs the evaluation of suitability of various alternatives" by saying that they directly do not make decisions that require an assessment of future alternatives, or that the assessment of alternatives is not currently meaningful due to the applicable legislation: *"(…) there are no such needs to consider future alternatives. Maybe it would be good, but at present there are no such needs"* (1); *"a complicated question, we [are working] according to documents and the legal acts—we do the way the legal acts provide"* (3); *"currently they are not related, practically, we actually work according to instructions, rules, planners have their visions, they improvise within the framework of the rules, they circumvent them, so to speak. But everything is restricted too much, and that attitude of specialists—jut propose anything new... [their reaction is negative]"* (6), *"[The modelling tool] needs not to be like a legal act, it should not be "sacred" in order not to have the only one correct answer"* (11); however, they acknowledge that they would be interested in an opportunity to evaluate the alternatives: *"For the evaluation of alternatives, it would be better to perform machine calculations [to avoid] the factor of subjectivity of human experts"* (4);*"You always evaluate [options]: be realistic, optimistic and pessimistic"* (9); *"We have certain alternatives to evaluate if we act like before, we will miss something in the future, if we do something differently, find land, prove the need for forest, we will implement the goals that Lithuania has assumed in relation to climate change. If we agree with the aggressive agricultural programme, we may not need those forests. Thus, we have alternatives everywhere"* (10); *You always have to consider several options"* (17). |

| Questions | Summary of the Responses from the Informants and Selected Citations (Hereafter, the Numbers in Parentheses Refer to the Corresponding Informants' Numbers in Table 1) |
|---|---|
| 3.4 To what extent and how could the decisions adopted by you be influenced or changed by a scientifically based tool that enables the modelling of various forest development alternatives?<br><br>3.4.1 What decisions in your opinion would it be easier to adopt if you had such a tool?<br><br>3.4.2 What information (data) in your opinion should such a modelling tool provide? | Concerning the development of a scientifically based tool which allows the modelling of different forest development alternatives, the opinions of informants ranged from completely pessimistic to optimistic ones. For example, for some, such a tool *"in theory is likely facilitate the work, but in practice it would complicate it"* (2). Some informants would agree with the benefit of such a tool, but they are not entirely sure about its unquestionable benefits: *"We apply modelling [alternatives] minimally, so the need exists"* (15); *"to a certain extent it would facilitate, it would be a support, you would not have to develop the [data] yourself. That would be a facilitation"* (16). Many informants would be very optimistic about the alternative modelling tool: *"If they were able to create such a versatile instrument, anyone would be happy, both the Department [Ministry of Environment] and SFE. But maybe [the modelling tool is just] a science-oriented tool. But would those models be closer to real modelling? If so, it would be really excellent"* (8); *"First, it would make things easier, but it would also add quality or objectivity. Now, although we consider the alternatives when making decisions, we propose to the extent [allowed by] the available information, subjective understanding. And [the options of alternatives] would be easier supported by arguments, presented to the public, more objective in terms of any decisions at national level"* (10); *"it would be much easier"* (12); *"It would be easier to decide, plan"* (13); *"it would change, accelerate and facilitate..."* (21). A certain scepticism regarding forest development alternatives modelling tool can be associated with the attitude towards the forestry scenario modelling in general and with depth of knowledge of this issue. Sometimes, it is feared that the modelling will not answer "all" questions: *"It is hard to expect for 100 percent forecast; inventory, growing models can be used to support many things. [Scenario modelling] would be undoubtedly helpful"* (17); *"If it [were] possible to weigh everything, if we agree that we can believe it, that would be great"* (18), or it makes little sense because there are few future alternatives and their cognition is of little significance: *"(...) Forecasts for the European forest sector—there are not many alternatives, it's all the same for the whole decade. Something might be updated, but in terms of its development, this is "business as usual", climate change, biodiversity, that's all, perhaps there might be four alternatives (...)"* (5).<br><br>3.4.1 In this paper, first, we attempt to relate scenario modelling to the support of the decisions made, i.e., with the decisions that will be made anyway and which will not be perfect in any case. Naturally, in order to communicate one or another methodological solution for scenario modelling, a vision of future forests is presented in simplified future conditions that can be easily imagined. From a practical point of view, only that tool for modelling various forest development alternatives is significant which preconditions the reduction of risk of error in decision-making and the reduction of the negative consequences of poor decision by choosing from several alternatives. Some informants also mentioned the problem of timeliness both in assessing alternatives and making decisions: *"Now we [decide] according to our own understanding, sometimes we order some scientific research. But then you lose time, it's a very important nuance. Having a pre-built system would add speed to your decisions. Now sometimes we get some results, but they stretch over time: if we need a scientific research, we lose a year"* (10); *"This is highly relevant. (...) we, our decisions should rely on that. But the system is not functioning. If a scientific order is made, it is not fast. If you make an inquiry, you can get a reply after half a year. [Scenario modelling] could change the decisions, we would be more expeditious. We could inform both the owners of private and public [forests] (...). To calculate "what if" scenarios, if climate changes"* (19).<br><br>However, all informants unanimously stated that system assessing alternatives would greatly facilitate the decisions, regardless of who makes those decisions (i e., in a sense, by distancing themselves from the informant's job functions): *"It's a very good tool for forest policy-makers as well as managers"* (3); *"could choose the alternative, what to do, whether to do at all, what pays off, what the potential income is"* (7).<br><br>3.4.2 However, when asked what information an alternative modelling tool should provide, the informants were quite modest: *"It depends on the type of the forecast made"* (19), or were unwilling to go into detail: *"dendrometric parameters, ecosystem services and economic matters"* (6). In principle, most of the informants have already answered this question when talking about the expectations of information on future forest resources. |

| Questions | Summary of the Responses from the Informants and Selected Citations (Hereafter, the Numbers in Parentheses Refer to the Corresponding Informants' Numbers in Table 1) |
|---|---|
| 3.5 To what extent and how could the decisions adopted by you be influenced or changed by a scientifically based tool that enables the optimization of decisions by evaluating the suitability of various alternatives? In other words, how does such a tool help answer the question 'what to do?' <br> 3.5.1 For what decisions would such a tool be useful? | Practically no specific answer was given to the question "To what extent and how the decisions adopted by you could be changed by a scientifically based tool that enables optimisation of the decisions adopted which performs the evaluation of suitability of various alternatives?", most probably due to the understanding of the content of the optimisation process itself: *"I cannot really imagine. I there is a tool that optimises all solutions, the need for science disappears. [It would be necessary] to use it as the starting point or to criticise it. And in practice, of course, it would affect and change something. If the tool shows one thing, and I show another, at least you will think"* (17). However, the informants–scientists noted that installation of such a tool could be problematic: *"This would be a complex tool, even if we are able to develop it. Probably it would be difficult to manage"* (19). Let us say that in Germany it is believed that such a tool is necessary; however, *"it is very difficult to program it. We are thinking about that but have not implemented it yet."* (20). |

Part 4. Other

| | |
|---|---|
| 4.1 What is the relevance of land use information (describing the past, present and future states) for you? | Experts with professional experience in the field of forestry were interviewed, so, by asking them questions about information on land use, its relevance to land use scenario modelling, and the tool for optimisation of related decisions, we were seeking to obtain additional information be relevant for the analysis of the land use scenario modelling system needs. In that regard, the majority of informants, in particular the ones directly related to forest agencies, were very careful. The land use scenario modelling information was mostly related to the history of the existing forests: *"From SFE's point of view, I cannot say that land use modelling would be very important, but it is interesting in terms of retrospective—there was a forest, there was no forest (...). The modelling should answer what is the benefit for certain land uses. It would certainly help to optimise land uses."* (8); *"There was a project where we deciphered land use from the time of the war, how forest coverage was changing. [Forest coverage] was changing more differently than we thought, the land was afforested in some places and deforested in another ones"* (18). The informants–foresters provided practically no specific proposals concerning the issue of land use scenario modelling and the functionality of the related solutions optimisation tool. In addition, this may be related to the view that the area of forest land cannot decrease, which means that there is nothing left for modelling at the same time: *"Forest land is sacred and it cannot be changed"* (5), unless we would like to predict the change of forest coverage: *"it would be relevant to forecast changes in forest coverage, but it is still mainly looked at the practical, current state or to the past"* (1); *"I as a private person [wonder] why not to let forest grow in certain areas of land? I have already done that and I can see the accumulation of carbon, wood, and, at the same, value. According to the ideas of private [owners], maybe it does not make sense to hold land, fertilise poor soil, if a person can see in [the land use scenario modelling tool] what values [the specific land use] can create, that would be fine"* (19). |
| 4.2 How relevant would it be for you to use the land use scenario modelling and related solutions optimization tool? What functionalities or features would you like to see in such a tool? | One of the informants talking about the experience of development and use of forestry scenario modelling systems identified the essential problem related to this issue: *"Forest scenario modelling is quite a difficult task, when we were preparing some scenarios, continuity was planned, but there were quite primitive attempts from the operating side in addition to other functions, but [such attempts] extinguished. If it [were] created, there would be a great deal of interest from users and science, especially in decision-making. In many places you could dot the i's and cross the t's without unnecessary dilettante discussions. We could express the desire to achieve this in any way"* (19). Thus, it is very difficult to foresee the entire functionality of a forestry scenario modelling system in advance. The system must be open to modernisation, just as attention must be ensured for such modernisation. In this regard, the proposal of another informant is of particular relevance: *"Make the decision makers to state that they really need it because they often say they do not need new research, so when you do, you will say they needed it"* (17). |

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
