# Peer review of "Forestry Scenario Modelling: Qualitative Analysis of User Needs in Lithuania"

_forests, doi:10.3390/f15030414_

Round 1

Reviewer 1 Report (Previous Reviewer 1)

Comments and Suggestions for Authors

MS can be accepted in present form

Author Response

Thanks for your evaluation and comments

Reviewer 2 Report (Previous Reviewer 4)

Comments and Suggestions for Authors

Round 2

Title: Forestry scenario modelling: qualitative analysis of user needs in Lithuania

Manuscript number: forests- 2867669 

I would like to thank the authors for addressing most of my previous comments. Now the manuscript is improved, though some issues still need attention.

  1. The abstract is extended, which includes more background concepts.
  2. In the introduction section, line number 123, the ‘term end users’. Do you mean those 11 institutions represented by 21 individuals?
  3. Line number 186, replace the 21 by words.
  4. Do you think that your study answers one of your objectives? “…our objective is also to address methodological issues related to specifying the requirements of modernized forestry DSSs." In your result part, there are no findings that answer this objective, but there are paragraphs in the discussion and conclusion sections. However, the concepts are general truths about qualitative and quantitative methods, which are not directly supported by your findings.

Author Response

Please, find our responses below: 

I would like to thank the authors for addressing most of my previous comments. Now the manuscript is improved, though some issues still need attention.

The abstract is extended, which includes more background concepts.

Our response: the abstract was shortened, removing some background information from the beginning. This adjustment does also refer to the comment by Reviewer 3.

In the introduction section, line number 123, the ‘term end users’. Do you mean those 11 institutions represented by 21 individuals?

Our response: We assume any potential user of a system which is under development. The sentence was modified, adding “supposed end users” and removing “… but will support their activities”.

Line number 186, replace the 21 by words.

Our response: Replaced

Do you think that your study answers one of your objectives? “…our objective is also to address methodological issues related to specifying the requirements of modernized forestry DSSs." In your result part, there are no findings that answer this objective, but there are paragraphs in the discussion and conclusion sections. However, the concepts are general truths about qualitative and quantitative methods, which are not directly supported by your findings.

Our response: We agree with the comment. The following text was inserted to the end of Discussion: “In principle, our findings regarding the needs and expectations of users of modernized forestry DSSs assume indirect validation of methodological questions through the specification of requirements using qualitative interviews. However, the results of this study have been utilized in planning real-life DSS implementations, thus confirming the methodological soundness of the approach.”

Reviewer 3 Report (New Reviewer)

Comments and Suggestions for Authors

Forests are vital for the environment and humans, providing oxygen, protecting soil and water, and harboring diverse species. They offer timber, recreation, and shape the climate and landscape, making them crucial for our well-being and survival. We must protect and care for forests to ensure their continued benefits for future generations.

The authors conducted a qualitative analysis of user needs for modeling forest scenarios in Lithuania. The purpose of the research is properly defined and the methodology is well described. The authors achieved the aim of the research and conducted a scientific discussion regarding the obtained results.
Below are some comments that will improve the quality of the article:

1. The abstract should be shortened (shorten and leave only the most important conclusions) 2. Define the concept of Forestry scenario modeling 3. In chapter 2.1. Study Area we have information about the forest cover in Lithuania. You should provide the date of the state of forest cover and it is worth writing down what plans Lithuania as a country has to increase the forest cover. This is one of the aspects of each country's spatial policy. The idea of Forestry scenario modeling is also new afforestation, so it is worth emphasizing what the assumptions are for the next 10-15 years.

It is worth showing how Lithuania compares to other countries in terms of forest cover and plans to increase forest cover (especially in the EU).

4. Literature citation should be corrected (e.g. line 69, should be: [6-12])

As the authors rightly pointed out, in Lithuania a large share of forests is owned by private persons. Their opinions were not taken into account in this research. It is worth continuing the undertaken research and learning about the needs of forest owners (private persons) in the field of forestry scenario modeling.

Author Response

The abstract should be shortened (shorten and leave only the most important conclusions)

Our response: the abstract was shortened, removing some background information from the beginning. This adjustment does also refer to the comment by Reviewer 2.

Define the concept of Forestry scenario modeling

Our response: The following sentences were added to the introduction: Forestry scenarios are supposed to depict and describe a range of possible, preferable and probable future developments of forests and forest management. They are implemented using diversity of computer-driven modelling tools.

In chapter 2.1.Study Area we have information about the forest cover in Lithuania.You should provide the date of the state of forest cover and it is worth writing down what plans Lithuania as a country has to increase the forest cover. This is one of the aspects of each country's spatial policy. The idea of Forestry scenario modeling is also new afforestation, so it is worth emphasizing what the assumptions are for the next 10-15 years.

Our response: Information on forest land proportion added.

It is worth showing how Lithuania compares to other countries in terms of forest cover and plans to increase forest cover (especially in the EU).

Our response: Unfortunately, the plans and solutions to increase the forest land proportion are much beyond our study. We have modelling results on the limits for forest land proportion in Lithuania, however, we would prefer not to use them in current study, also bearing in mind large number of own citations

Literature citation should be corrected (e.g. line 69, should be: [6-12])

Our response: Corrected

As the authors rightly pointed out, in Lithuania a large share of forests is owned by private persons. Their opinions were not taken into account in this research. It is worth continuing the undertaken research and learning about the needs of forest owners (private persons) in the field of forestry scenario modeling.

Our response: The opinions of private forest owners were represented by the informants from Forest and Land Owners Association of Lithuania and one private forest company. Indirectly, some informants owned and managed their own forest land.

This manuscript is a resubmission of an earlier submission. The following is a list of the peer review reports and author responses from that submission.

Round 1

Reviewer 1 Report

Comments and Suggestions for Authors

The topic of the paper is interesting and more studies regarding forest management scenario modelling in forestry sector are needed. However, the content and scientific soundness of the paper is very vague. I am not sure what key messages the authors try to provide from this manuscript. Also it is not clear what additional knowledge this manuscript gives to the scientific communities and what is the added value of this paper. If we review it from academic paper, it still need to be improved much.

Specific comments:

The introduction lacks a deeper theoretical background to the issue. More intense engagement with recent related literatures is encouraged in positioning the study and justifying its aim. Moreover, description of the chosen research method should be included within the methodology part instead of the introduction.

The methodology is unclear:

    - there is no clear information about the statistical representativeness of the given sample of respondents. Are 11 institutions really enough?

-   - how (based on which parameters) were the (first) companies/respondents selected for in-depth interviews?

-     - it is quite strange to interview several people from one institution. Why did you proceed this way?

It is obvious from the scope of the questionnaire that the authors collected a large amount of useful data. However, the interpretation of the results is not scientific and the formulations are written like a story, which is not appropriate for a scientific journal. For the data analysis, I do not see strong analytical tools used in the manuscript, the authors only transcribed the interviews. There is a lack of evaluation of the achieved results and possible recommendations.

Discussion part has absolutely no character of a discussion, it is more like summary. The results should be discussed with other scientific findings and works. Table 2 does not belong to the discussion but to the results.

This paper need to be improved by theory analysis and academic discussion, and also by statistical analysis based on enough sampling. Therefore, I cannot recommend the paper for publication.

Author Response

Response to Reviewer 1 Comments

The topic of the paper is interesting and more studies regarding forest management scenario modelling in forestry sector are needed. However, the content and scientific soundness of the paper is very vague. I am not sure what key messages the authors try to provide from this manuscript.

Response: We accept the weak scientific soundness of the manuscript. We tried to strengthen the scientific message of the case study in the abstract, introduction and discussion, emphasizing social benefits of the study related to design and implementation of forest DSSs and the role of quantitative research. The major trends have often been that the forest DSS are adopted to changing conditions, however, here we deal with attempt to go different direction.

Also it is not clear what additional knowledge this manuscript gives to the scientific communities and what is the added value of this paper. If we review it from academic paper, it still need to be improved much.

Response: We tried to formulate the scientific message more clearly (see the comment above). First of all, we aim to demonstrate the importance of qualitative approach in exploring the phenomenon which is characterised using intangible attributes

Specific comments:

The introduction lacks a deeper theoretical background to the issue. More intense engagement with recent related literature is encouraged in positioning the study and justifying its aim. Moreover, a description of the chosen research method should be included within the methodology part instead of the introduction.

Response: The introduction was modified with a more elaborated research problem. The description of methodological aspects was removed from the introduction. We updated the methodology, however, the explanation of our choices was moved to the completely revised section of the Discussion

The methodology is unclear:

    - there is no clear information about the statistical representativeness of the given sample of respondents. Are 11 institutions really enough?

Response: We provided explanations on the sample size in the Discussion, emphasizing the differences of qualitative and quantitative approaches. Qualitative approach limits the statistical inference to other populations, such as private forest owners. However, it is widely accepted that the focus in qualitative research is not on the sample size but on collecting detailed information about the individuals under analysis. Extra explanations regarding the choice of institutions were included in the updated version of the manuscript. We covered the most powerful actors in the field. Increasing the number of informants (contrary to the number of respondents in the quantitative study) may introduce redundancy in data. 

-   - how (based on which parameters) were the (first) companies/respondents selected for in-depth interviews?          

Response: The description of selecting informants for in-depth interviews was adjusted in the updated version of the manuscript. We have a special paragraph with methodological requirements for selecting the informants followed by an explanation of our approaches used. 

-     - it is quite strange to interview several people from one institution. Why did you proceed this way?          

Response:  We do not see anything strange, because the functions and competencies of different experts within the same institution may be radically different. E.g., the State forest service covers all aspects of controlling the implementation of forest policy in the country with the departments completely different in tasks and the background of personnel. We tried to explain this issue in the updated version of the manuscript. The interviews were structured in a way, that each informant was asked on whether he or she managed to cover all aspects relevant for the study and referring to a specific institution, asking also to identify the names of additional experts. 

It is obvious from the scope of the questionnaire that the authors collected a large amount of useful data. However, the interpretation of the results is not scientific and the formulations are written like a story, which is not appropriate for a scientific journal. For the data analysis, I do not see strong analytical tools used in the manuscript, the authors only transcribed the interviews. There is a lack of evaluation of the achieved results and possible recommendations.       

Response: We restructured the introduction of the results. The summary of results, based on citations of selected parts from the transcriptions, was moved to the annex. This we suppose are an essential part of report of qualitative study and we used a search of repetitions and contradictions, identification and refinement of ideas relevant to the study, and summarising and description of results obtained. Transcriptions are available but they are not provided with the paper, as they are in Lithuanian language. We did not use specialised software solutions, like Atlas.ti or MAXQDA. We suppose, they are efficient when there is a need to analyse huge amounts of data, which was not a case in our study. However, our approach has proven to be successful was to store the data in well planned Excel worksheets and use standard search functionality. Another point is that we have always used two persons to conducts the interviews, meaning, that one researcher has always been following the conversation and structuring the further analyses in advance. Another disadvantage of using computer programs is that they go beyond their cost. Important factor, limiting the use of specialised package also the interviews, done in Lithuanian language.

The discussion part has absolutely no character of a discussion, it is more like summary. The results should be discussed with other scientific findings and works. Table 2 does not belong to the discussion but to the results.

Response: The discussion was rewritten, elaborating new version of the results based on Table 2 and associated text. Discussion was completely rewritten focusing both on the issues of qualitative vs quantitative research and the other studies of user needs in the field of forest DSSs.

This paper need to be improved by theory analysis and academic discussion, and also by statistical analysis based on enough sampling. Therefore, I cannot recommend the paper for publication.           

Response: We accept this evaluation and are thankful for the comments. 

Reviewer 2 Report

Comments and Suggestions for Authors

In general, the manuscript is interesting and relevant to the journal but needs major revisions to be published in this Journal. More and detailed comments are provided in the manuscript file.

Comments on the Quality of English Language

Minor editing of English language required

Author Response

Response to Reviewer 2 Comments

In my opinion, this title is very general and if possible, it can be more clearly stated          

Response: There were minor changes done in the title. However, we would like to keep the focus on qualitative methods in the title, hopefully, increasing scientific value of, in principle, a case study 

The necessity of this investigation should be summarized in one sentence

The result of this research is stated in only one sentence and in a very general way

While it is necessary to express the main quantitative and qualitative results of the research in the abstract

Response: The abstract was modified as were also other parts of the manuscript. We hope that we managed to follow your recommendation, by adding more on the results and emphasising the focus of our study on the qualitative methods

 It is better to discuss the scientific and operational achievements of this research based on its results in a clearer sentence

Response: We hope that major modifications of texts removed this issue

You can use the same sentences in the title of the paper and in the text

Forestry scenario modeling

OR

Forest management scenario          

 Response: Thanks for the comment. We hope the forest management scenario disappeared. We use the term of forest management scenario in other context, which is not relevant for current study

The sentence needs to be corrected

Response: This sentence was rewritten. 

Cite the reference          

Response: The citation was adjusted. Note, that the sentence was changed

Cite the reference          

Response: The citation included

After stating the goal, state its practical and operational achievements

Response: The formulation of the goal was changed. We have also added some statement on the practical importance of the study. Identification of practical achievements we also provide in the updated version of the Discussion and Conclusions

In addition to the land cover, the general climate of the region can also be mentioned

Response: The sentence “Lithuanian forests belong to the European hemi-boreal mixed broadleaved-coniferous forest type in the transitional zone between the boreal coniferous and the nemoral broadleaved forests” was inserted

If possible, you can also mention the role of forestry in land management and its place in the country's economy

Response: The following text was inserted: “Throughout the history of Lithuania, forests have consistently held a crucial position in the country's economy. This significance stems from the fact that wood, being one of the limited domestically accessible raw materials, plays a fundamental role. The importance of other forest functions, such as biodiversity conservation, carbon sequestration, and recreational use, has always been significant”. Please, note, that we elaborate on the forest management system later, in the rewritten version of the Discussion

You can briefly refer to the history of forestry plans in the study area in the background            

Response:  We could not fully understand the comment. The forestry planning, especially as it regards the use of DSS, is briefly discussed in the Discussion chapter

Explain more about these analyses         

Response: The methodology chapter was modified. We tried to describe step-by-step all components of the study. Please, note, that some methodological aspects are also discussed in the new version of the Discussion

Wasn't there a need to conduct significance analysis and more statistical analysis from this survey?            

Response: The focus of the current study was on using qualitative methods as an alternative to quantitative surveys. Therefore, we may not provide statistical tests as assumed in conventional forestry research. We provided explanations on the sample size in the Discussion, emphasising the differences of qualitative and quantitative approaches. The qualitative approach limits the statistical inference to other populations, such as private forest owners. However, it is widely accepted that the focus in qualitative research is not on the sample size but on collecting detailed information about the individuals under analysis. Extra explanations regarding the choice of institutions were included in the updated version of the manuscript.

The results of this research and how to present it should be revised

The detailed description of the topics seems boring and it is better for the authors of the paper to use tables and figures as much as possible to show the results better.      

Response: The way the results are introduced in the adjusted version of the manuscript was completely changed. We moved the most boring part into the annexes. We may not avoid providing the most important part of qualitative research, which is an output from the analysing of transcriptions (corresponding to the data table of conventional forestry study). The new version of the results is based on the previous Discussion

While focusing on the main results of this research, this section should compare these results with the results of previous related studies          

Response: The discussion chapter was rewritten and now is covering a brief interpretation of our results but with a focus on qualitative methods vs quantitative surveys and on the DSS user needs studies. 

Reviewer 3 Report

Comments and Suggestions for Authors

Introduction Section: I would suggest making significant changes. First, avoid grey literature (i.e., reference 1). Second, some statements are not supported by data or bibliographical references, which may lead to being considered value judgments. Third, in order to explain critical questions about the methodology used, grey literature is not the best option (i.e., reference 16)

A reader does understand why the main weakness of a quantitative method is the preparation of a questionnaire, and the authors implement a questionnaire for their qualitative analysis.

The methodology has not been explained in detail: no bibliographic references and no description of the main characteristics (i.e., snowball technique). In short, as I have said before, a reader cannot replicate this analysis.

Why did the authors not use software focused on qualitative research (i.e., Atlas.ti or a similar one).

The Discussion Section lacks a comparison of the results obtained and/or the methodologies used with other papers. This is not advisable in a scientific article.

The Conclusions Section must be rewritten, due it is not well linked with the manuscript’s primary objective (p. 3).

Author Response

Response to Reviewer 3 Comments

Introduction Section: I would suggest making significant changes. First, avoid grey literature (i.e., reference 1).           

Response:  Well, we hope we understood what was meant under “gray literature”. The references were modified essentially, including reference 1. The modifications included both removing old and adding new references

Second, some statements are not supported by data or bibliographical references, which may lead to being considered value judgments.

Response:  We hope the issue disappeared in the adjusted version of the manuscript.

Third, in order to explain critical questions about the methodology used, grey literature is not the best option (i.e., reference 16)

Response:  The methodology was adjusted step-by-step introducing the stages of planning, conducting and analysing the interviews and explaining the specificity in context of current study. We have included a paragraph in the new version of the Discussion dealing with methodological aspects of qualitative studies. We would like to keep the reference previously identified as 16, at it elaborates on methodological aspects of qualitative methods, but first of all it demonstrate the use of qualitative approach in the field dominance of quantitative surveys. Nevertheless, referring to the forest owner classifications was removed.

A reader does understand why the main weakness of a quantitative method is the preparation of a questionnaire, and the authors implement a questionnaire for their qualitative analysis.               

Response:  We developed a completely new version of a Discussion where we elaborate on qualitative methods vs quantitative. Yes, we used for our purposes “questionnaires” with open-ended questions to elicit detailed responses from informants, however, they were not displayed to informants

The methodology has not been explained in detail: no bibliographic references and no description of the main characteristics (i.e., snowball technique). In short, as I have said before, a reader cannot replicate this analysis.

Response:  The methodology was adjusted, step-by-step introducing the study. Adjusted version of the discussion is supposed to deal with the explanation of the methodology and listing the advantages and limitations of our approach

Why did the authors not use software focused on qualitative research (i.e., Atlas.ti or a similar one).    

Response:  We accept that there are software solutions to explore the materials of in-depth interviews, like Atlas.ti or MAXQDA. However, they are efficient when there is a need to analyse huge amounts of data, which was not the case in our study. However, our approach has proven to be successful was to store the data in well planned Excel worksheets and use standard search functionality. Another point is that we have always used two persons to conduct the interviews, meaning, that one researcher has always been following the conversation and structuring the further analyses in advance. Another disadvantage of using computer programs is that they go beyond their cost. The important factor, limiting the use of specialised package also the interviews, done in the Lithuanian language.

The Discussion Section lacks a comparison of the results obtained and/or the methodologies used with other papers. This is not advisable in a scientific article.            

Response:  The Discussion was rewritten. Hopefully, it goes in line with the standards for scientific discussions

The Conclusions Section must be rewritten, due it is not well linked with the manuscript’s primary objective (p. 3).

Response:  The Conclusions were joined to the Discussion, they were essentially modified as was also the formulation of objectives. 

Reviewer 4 Report

Comments and Suggestions for Authors

Comments on forests-2409144-peerr-reveiw

Title: Forest management scenario modelling: qualitative analysis of user needs in Lithuania

Over all comments

Overall comment: The manuscript structure is fine, the write-up is clear-to the standard.

The study will have substantial contribution to the field and the authors’ raised an interesting topics and used clear methodology. The quality and clarity of the writing is good, however the following issues needs attentions.

Abstract

i. The abstract is well done but lacks clearly indicating how 21 samples were selected from the target groups.

1. Introduction

This section is well organized, but the author should address the following points:

i. The introductory concept on the term, ‘forest management’ scenario modeling’

ii. On you objective, “ … to explore the needs of the potential users of forestry scenario system in Lithuania”. From this sentence, what I understand is that the authors are interested to assess the potential users of the scenario. My interest here is that, is it possible to know that using 21 sample?

In addition, how do you decided to include 21 from 11 institutions? What was the total experts from the 11 institutions? Is there only 11 institutions in Lithuania?

2. Method section

i. On the description of the study area, there is no citations.

ii. No clear step on the technique of selecting the 21 sample/informants. The procedure should be scientific. For unknown you used snow ball, but how did you handle for others?

3. Results

The result section is descriptive and the authors stated the results using quotation plus italics form. Is it normal to use both?

On the statements, there are numbers in parenthesis, which refers the sample supporting the ideas from the total, which is 21. My interest here is that, what are the relevance to state the ideas that was supported by aa few like, 2,3,4,5,6,7, up to halves of the sample?

4. Discussion section

The existed information is good but the authors should explicitly indicate for the policy-makers, which forest scenario modeling is relevant and the scale-at plot or national level.

Consider if related studies are available and discuss with your findings.

Table 2 is part of study result and possible to be in the results section.

4. Conclusions

It is based on the study results and no comment.

In addition, see the pdf version for a few editorial comments.

Author Response

Response to Reviewer 4 Comments

Title: Forest management scenario modelling: qualitative analysis of user needs in Lithuania

Overall comments

Overall comment: The manuscript structure is fine, the write-up is clear-to the standard.               

Nothing to respond

The study will have substantial contribution to the field and the authors’ raised an interesting topics and used clear methodology. The quality and clarity of the writing is good, however the following issues needs attentions.

Nothing to respond

Abstract

  1. The abstract is well done but lacks clearly indicating how 21 samples were selected from the target groups.

Response: The abstract was rewritten. The part on the samples is formulated as follows: “A total of 21 informants from 11 different institutions, which hold significant power and expertise in forest decision-making, were interviewed. The purpose of these interviews was to gather their perspectives on the potential forest decision support system in the country, aiming to address the majority of their needs”. The motivation behind the sample is explained both in Methodology and Discussion (adjusted versions)

  1. Introduction

This section is well organized, but the author should address the following points:

  1. The introductory concept on the term, ‘forest management’ scenario modeling’

Response: We removed the “forest management scenario”. We also inserted a text explaining scenario modelling, decision support systems and the issues leading to the need of current study.

  1. On you objective, “ … to explore the needs of the potential users of forestry scenario system in Lithuania”. From this sentence, what I understand is that the authors are interested to assess the potential users of the scenario. My interest here is that, is it possible to know that using 21 sample?

Response: We provided the explanations on the sample size in the Discussion, emphasising the differences of qualitative and quantitative approaches. Qualitative approach limits the statistical inference to other populations, such as private forest owners. However, it is widely accepted that the focus in qualitative research is not on the sample size but on collecting detailed information about the individuals under analysis. Extra explanations regarding the choice of institutions were included in the updated version of the manuscript. We covered most powerful actors in the field. Increasing the number of informants (contrary to the number of respondents in the quantitative study) may introduce the redundancy in data.

In addition, how do you decided to include 21 from 11 institutions? What was the total experts from the 11 institutions? Is there only 11 institutions in Lithuania?           

Response: The above response refers to this comment, too. We explained the choice of institutions in abstract, methodology and the discussion. Our approach was to cover all actors of Lithuanian forest policy with a power to influence the solutions behind forest DSS. Increasing the sample size and increasing the number of informants was not expected to increase the extent of information collected.

2. Method section

  1. On the description of the study area, there is no citations.

Response: The description of study area was enrichened with the citations of some documents describing the forest and forestry. However, we have no references for the geographic location. We inserted among the references the Lithuanian Statistical Yearbook of Forestry, which provides also basic facts on geographic location, too

  1. No clear step on the technique of selecting the 21 sample/informants. The procedure should be scientific. For unknown you used snow ball, but how did you handle for others?

Response: We adjusted the methodology, emphasising also the technique for selecting the informants. Briefly, politically most weighty and best informed due to their positions informants were selected first. They suggested additional informants and this was repeated with other informants until we started to receive suggestions with already interviewed candidates

  1. Results

The result section is descriptive and the authors stated the results using quotation plus italics form. Is it normal to use both?  

Response: The results were changed. However, the previous version of the results was moved to annexes as we suppose the summary of responses is an important part of qualitative research. We used italic text intentionally, as it cites the informants. We accept, that this is a translation from Lithuanian in most cases and there is no opportunity to provide the non-edited texts.

On the statements, there are numbers in parenthesis, which refers the sample supporting the ideas from the total, which is 21. My interest here is that, what are the relevance to state the ideas that was supported by aa few like, 2,3,4,5,6,7, up to halves of the sample?

Response: We explained in the text that the numbers refer to the ID of informant in Table 1. 

  1. Discussion section

The existed information is good but the authors should explicitly indicate for the policy-makers, which forest scenario modeling is relevant and the scale-at plot or national level.               

Response: The discussion was completely rewritten. The survey was focused on the use of DSS for strategic and tactical planning both at national and estate levels. The scale effect was also questioned during the interviews.

Consider if related studies are available and discuss with your findings.   

Response: The updated version of the discussion covers comparisons of our results with the findings of other authors. More specifically, we focus on the methodological aspects (qualitative vs quantitative) and the surveys of DSS needs. 

Table 2 is part of study result and possible to be in the results section.    

Response: Absolutely agree. New version of the results is based on Table 2. 

  1. Conclusions

It is based on the study results and no comment.

Response: Conclusions were merged with the discussion. We also reformulated the conclusions to look more different from condensed results. 

In addition, see the pdf version for a few editorial comments.

Include sources/citations            

Response: Inserted, see the comment above

one word

Response: corrected

Why only 21? was experience the criteria you applied to selecet your samples?

Response: Commented also above: We provided the explanations on the sample size in the Discussion, emphasising the differences of qualitative and quantitative approaches. Qualitative approach limits the statistical inference to other populations, such as private forest owners. However, it is widely accepted that the focus in qualitative research is not on the sample size but on collecting detailed information about the individuals under analysis. Extra explanations regarding the choice of institutions were included in the updated version of the manuscript. We covered most powerful actors in the field. Increasing the number of informants (contrary to the number of respondents in the quantitative study) may introduce the redundancy in data.

should be a continous sentence

Response: Corrected

Use in word

Response: Corrected “six”

Use the full term before using the abbreviation

Response: Adjusted: European Forestry Dynamics Model

Belwo the table, write the full term of the abbreviations-name of the organizations.

Response: Full names were provided in the table itself

What the numbers at the end of the quate sentence refers? Is it from the total sample size? If so why the sum is larger thatn 21? Or the respondents can give more than one answers? If so, in the method section, you have to indicae that the repondents can give more than one answer for some of the questions.

Response: Responded above: We explained in the text that the numbers refer to the ID of informant in Table 1.

why in future tense form? the activitiy is alre\ad done and should be past tense form.     

Response: This sentence disappeared in the adjusted version of the manuscript

This can be move to the results section

Response: Done

Round 2

Reviewer 1 Report

Comments and Suggestions for Authors

Congratulations to the authors that they have managed to significantly  improve the paper. It is obvious, they did a lot of work on the MS. However, there are still some parts, that should be improved: 

- the results are very briefly described - the result part is the most important part after all, and in this paper it receives the least attention.

- I consider the discussion to be the weak part of the paper, too. A lot of attention is paid here to the discussion of selected methodological procedures - some parts would be more appropriate for the methodology part.

Author Response

Response to Reviewer 1 Comments

Congratulations to the authors that they have managed to significantly  improve the paper. It is obvious, they did a lot of work on the MS. However, there are still some parts, that should be improved:  

- the results are very briefly described - the result part is the most important part after all, and in this paper it receives the least attention. 

Our response: The Results chapter underwent the following modifications: First, we added several lines to introduce the results and guide the reader towards Annex 2 for a more detailed listing of the results, including quotes. We also expanded the Results elaborating on the specific interests of different groups of potential users of the forestry decision support system. This involved adding several additional paragraphs that introduced the information originally presented in Table 2. As a result, adjustments were made also to the Discussion section. 

- I consider the discussion to be the weak part of the paper, too. A lot of attention is paid here to the discussion of selected methodological procedures - some parts would be more appropriate for the methodology part. 

Our response: We made further improvements to the discussion section. Firstly, we made slight adjustments to the opening paragraphs of the discussion. We believe it is important to retain the explanations of our chosen methodological approach, specifically the use of a qualitative study based on in-depth interviews to identify the needs of future users of the information system, within the discussion. We feel that there are certain still unresolved issues that restrict introducing our approach as a methodological foundation in the Methodology chapter. We aimed to demonstrate the potential of using qualitative research for conceptualizing information systems, while acknowledging that certain aspects are still open to discussion. Additionally, we addressed specific research conditions that are locally specific and required explanation within the discussion. As a result, we would like to retain the paragraphs discussing the pros and cons of our methodological approach in the Discussion section. 

In addition, we have updated the discussion to incorporate the newly inserted messages from the Results section. The new paragraph focuses on the major direction of the implementation strategy, based on our findings and common practices. 

Reviewer 2 Report

Comments and Suggestions for Authors

 the manuscript is accepted for publication in its present form with no revisions

Author Response

the manuscript is accepted for publication in its present form with no revisions.
Our response: Thanks for the reviews and the opportunity for us to improve the manuscript.